# The barriers and enablers of outbreak reporting in the Asia-Pacific region: A mixed methods study of field epidemiologists

Amish Talwar[1]*, Matthew M. Griffith[1], Rebecca Katz[2], Martyn D. Kirk[1], Tambri Housen[1,3]

**1** National Centre for Epidemiology and Population Health, Australian National University, Canberra, Australian Capital Territory, Australia, **2** Center for Global Health Science and Security, Georgetown University, Washington, District of Columbia, United States of America, **3** School of Medicine and Public Health, University of Newcastle, Newcastle, New South Wales, Australia

* amish.talwar@anu.edu.au

## Abstract

Outbreak reporting is thought to be subject to technical, or capacity, barriers that physically hinder reporting and non-technical barriers, such as the threat of economic losses, that discourage governments from reporting. While technical barriers to reporting have been extensively explored, there is less evidence for the effects of non-technical barriers, particularly at the subnational level. To better understand the impact of both technical and non-technical barriers at the subnational level, we conducted an explanatory sequential mixed methods study involving field epidemiology training program (FETP) trainees and graduates in the Asia-Pacific region. We first surveyed study participants on the importance of putative outbreak reporting barriers and enablers using a three-point Likert scale and free text responses. We then interviewed respondents to more deeply explore key factors affecting outbreak reporting that were elicited from the survey. We calculated counts and percentages for all survey variables, and we coded free text responses from the survey and the interview transcripts. We then used thematic analysis to derive common themes from our findings. Fifty-six FETP trainees or graduates completed the survey, representing 19 Asia-Pacific countries and territories; of these respondents, we interviewed 16. The barrier noted by most survey respondents to have a high reporting impact was lack of staff (28, 50.0%). The enablers noted by most survey respondents to have a high reporting impact were training about what and how to report an outbreak (40, 71.4%) and sufficient surveillance resources (40, 71.4%). Based on our survey and interview findings, we elicited four thematic barriers – lack of capacity, behavioural barriers, political and socioeconomic barriers, and bureaucratic barriers – and three thematic enablers – building capacity, building a culture of reporting, and political and bureaucratic support. Although lack of capacity was found to disrupt reporting at the subnational level, several non-technical barriers also impacted reporting, particularly

**Data availability statement:** Although we have taken great effort to prevent participant identification in the data used for our study, there remains a possibility of inadvertent identification. Given the highly sensitive nature of the information the participants disclosed, inadvertent identification might be particularly harmful. To protect participants from any harm related to inadvertent identification and linkage with sensitive information elicited through this study, the authors decided that all data arising from this study will be retained at the Australian National University for at least five years following publications arising from the research. After this storage period, participant data will be fully de-identified and archived at the Australian Data Archive (www.ada.edu.au) for use by other researchers. In the interim, de-identified data can be made available upon reasonable request of the corresponding author with approval of the Australian National University Human Research Ethics Committee (human.ethics.officer@anu.edu.au).

**Funding:** The authors received no specific funding for this work.

**Competing interests:** The authors have declared that no competing interests exist.

economic and reputational concerns. Future studies should explore these barriers beyond the Asia-Pacific region.

## Introduction

In the wake of the COVID-19 pandemic, efforts to prevent the next pandemic have taken on new urgency. At the global level, World Health Organization (WHO) Member States have agreed to amendments to the International Health Regulations (IHR) and a new pandemic agreement [1,2]. These reforms aim to improve Member States' ability to detect, report, and respond to public health emergencies. The process of outbreak reporting has become a key area of focus, with the increasingly adopted 7-1-7 target recommending outbreak detection within seven days, notification of public health authorities within one day, and initiation of response activities within seven days after receiving the report [3]. However, delays in outbreak reporting at various levels of the reporting chain have persisted even after the revised IHR came into effect. This was the case for the Ebola outbreak in Western Africa and most recently during COVID-19, where local delays in reporting the existence and extent of the COVID-19 outbreak at its initial stages delayed the global response to the outbreak by several weeks, allowing it to grow into a global pandemic [2,3]. To prevent future pandemics, it is crucial to understand why nearly 20 years following the adoption of the most recent version of the IHR countries still experience failures reporting outbreaks in line with their international obligations.

States' failure to comply with IHR reporting obligations is typically ascribed to two main reasons: a lack of capacity to meet the obligations (i.e., the means to report) or a choice to not comply in part because the costs of reporting (such as loss of trade, travel, or resources like livestock) outweigh the potential benefits (i.e., the will to report) [4,5]. Based on our scoping review of the evidence for the putative factors affecting outbreak reporting, we found that much research has focused on improving countries' technical capacity to report public health events, including appropriate infrastructure, personnel, and funding; however, comparatively less research has examined non-technical reporting barriers and enablers, including socioeconomic, political, bureaucratic, and behavioural factors, that respectively discourage or encourage timely notification, particularly at the subnational level [6]. Multinational studies of the factors impacting outbreak reporting have had a similarly limited scope. For example, a study evaluating the successes and challenges in implementing the 7-1-7 target for outbreak detection, notification, and response in five countries in sub-Saharan Africa and South America examined the impact of technical capacity on outbreak notifications but only examined the impact of intersectoral coordination on reporting activities [7]. Furthermore, there is a lack of similar studies examining the barriers and enablers to outbreak reporting in the Asia-Pacific region, which has been the origin of several major outbreaks, including Severe Acute Respiratory Syndrome (SARS) and COVID-19 [8].

**PLOS Global Public Health**

To better understand the technical and non-technical barriers and enablers of outbreak reporting at the subnational level across different reporting environments in the Asia-Pacific region, we conducted a mixed-methods study on the knowledge, perspectives, and experiences of field epidemiology training program (FETP) trainees and graduates in the Asia-Pacific region using a survey and semi-structured interviews. By focusing on FETP-trained personnel, we investigated these barriers and enablers among officials who have extensively trained in and performed outbreak surveillance and notification. The results of this study can inform crucial improvements to outbreak reporting, facilitating faster outbreak response and ultimately preventing outbreaks from becoming pandemics.

## Methods

### Ethics statement

This study was conducted in accordance with the Australian National Health and Medical Research Council's National Statement on Ethical Conduct in Human Research. Written informed consent was obtained from all participants to participate in this study. The Australian National University Human Research Ethics Committee approved this study (Protocol 2023/196).

### Study design

We used an explanatory sequential mixed methods design (quantitative followed by qualitative phase) to assess the barriers and enablers of outbreak reporting (Fig 1, see S1 Text for study protocol) [8]. We grounded our approach in a utilitarian, pragmatic research paradigm in which our choice of methods was defined by our study's purpose – to examine "real-world" problems relating to outbreak reporting and to identify relevant solutions [9,10]. To do so, it was necessary to first quantify the perceived barriers and enablers to outbreak reporting to assess their relative importance and to then qualitatively explore the underlying contextual and experiential factors influencing these perceptions, enabling a more comprehensive and nuanced understanding than either method alone could provide. Accordingly, the data were connected with the quantitative phase informing the qualitative phase. Specifically, we administered a survey (including Likert scale questions and free text responses) to FETP trainees and graduates in the Asia-Pacific region to ascertain their views on the relative importance of putative outbreak notification barriers and enablers, followed by interviews with a subset of respondents to further explore their unique knowledge, perspectives, and experiences. Findings from the quantitative survey directly informed the development of a semi-structured interview guide for use during the interview phase, enabling deeper exploration of the identified barriers and enablers [11]. Thus, the qualitative findings built on and contextualized our quantitative findings, providing insights that would not have been realised had we exclusively used a quantitative or qualitative approach [12].

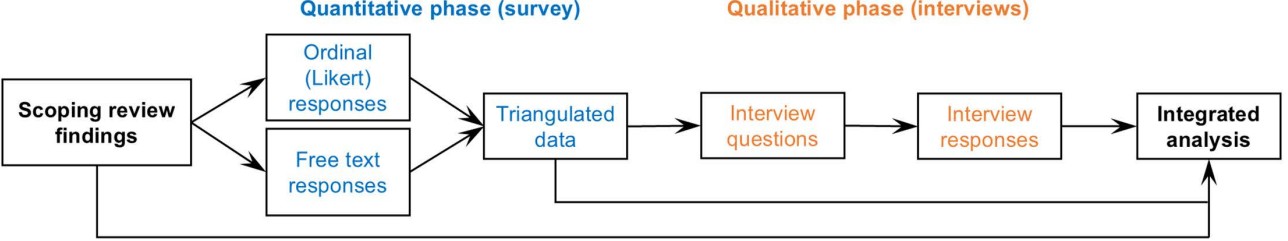

**Fig 1. Explanatory sequential study design. Adapted from Talwar, et al. (2024) [8].**

## Study recruitment and data collection

We aimed to obtain a convenience sample of study participants covering World Bank lower-middle income, upper-middle income, and high-income countries within the Asia-Pacific region by contacting 27 Asia-Pacific FETPs and regional FETP networks to distribute an invitation to participate in our study among their FETP trainees and graduates [13]. By recruiting among FETPs in Asia-Pacific countries with varying development levels, we hoped to maximize representation among a variety of geographic and economic settings. However, we were unable to solicit participation from Afghanistan and North Korea (the Asia-Pacific region's only countries classified as low-income) because of the absence of FETPs in these countries. We elected to recruit in this manner (instead of purposive sampling of specific trainees and graduates) because the programs themselves were the best source for FETP trainee and graduate contact information; furthermore, it is best practice to contact and work with program directors when seeking to access the FETP community for research purposes. We asked FETP Directors and FETP network staff to send an email describing the study and a link to the survey to their current trainees and graduates, who were free to either take the survey or decline. The recruitment period ran from 15 November 2023–30 August 2024.

The survey included demographic and background questions (age, gender, level of education, country of work, public health role, type of FETP training, and country through which the FETP training was completed) and questions on a three-point Likert scale ("High impact", "Some impact", and "No impact") to assess the relative importance of various putative barriers and enablers of outbreak reporting elicited during a previous scoping review of the literature [6]. The Likert scale questions related to capacity, coordination, and communication; training and socialization around reporting; motivation and incentives to report; authority to report; and whether reporting officials have felt pressure to not report or feared economic consequences from reporting (see S2 Text for copy of survey). Respondents also had the opportunity to further contextualize their responses using free text fields. Participation in the survey was anonymous, and no identifying information was collected unless participants voluntarily provided their contact details to be invited for a follow-up interview. We piloted the survey with four subject matter experts experienced with FETPs and outbreak reporting in the Asia-Pacific region.

For the semi-structured interviews, we employed a nested sampling design where survey participants were invited at the end of the survey to participate in an interview to further explore contextual realities and nuances related to identified barriers and enablers to outbreak reporting [14]. To maximize representation across the region and diversity of viewpoints, we invited all persons who indicated interest in being interviewed to participate. We sent each interested participant a Participant Information Sheet to review and a consent form to complete and return. The interview questions aimed to further examine barriers and enablers that stood out among the survey responses to more deeply explore how they presented within different reporting environments (see S3 Text for interview guide). Given the sensitivity of the topics to be discussed, we anonymised the interviewees for publication purposes. We piloted the semi-structured interview guide with a subject matter expert experienced with FETPs and outbreak reporting in the Asia-Pacific region. One author (AT) conducted interviews over Zoom. Interviews were audio recorded and subsequently transcribed verbatim and anonymised. We sent each transcript to the relevant interviewee to ensure accuracy and to mark for redaction information that could lead to inadvertent identification of the interviewee, in line with established practices of member checking and participant validation to enhance the credibility and trustworthiness of qualitative data [15]. We conducted all surveys and interviews in English.

## Data analysis

Consistent with our sampling methodology, we conducted a descriptive analysis of quantitative data, reporting counts and percentages for all variables. We then used the survey findings to construct the semi-structured interview guide, ensuring that the qualitative phase built directly on key issues identified in the quantitative data. One author (AT) inductively developed the coding scheme used for the analysis of qualitative data from the open-text survey responses in NVivo 14. Two authors (AT and MMG) subsequently refined this scheme to incorporate the richer detail and additional themes emerging

from the interviews, ensuring consistency while remaining flexible to the nuances of the new data (see S4 Text for final coding scheme). We merged the qualitative data from the open-ended survey questions and interview transcripts for the final analysis. While one author (AT) conducted all coding, another author (MMG) cross-checked coding validity by independently coding five transcripts to assess coding consistency. We calculated Cohen's kappa to determine the level of agreement between the coders. Coding comparison yielded a Cohen's kappa of 0.71, indicating substantial inter-coder agreement [16]. We then used thematic analysis to derive themes summarizing the coded data [17]. Data collection concluded when no additional codes emerged from the data (i.e., saturation) [18]; on our analysis of the interview findings, saturation was reached at 15 interviews. We described each interviewee's work setting, location of work according to World Bank region (in lieu of country to protect from inadvertent identification), and World Bank country income level classification [13]. As part of the merging process of quantitative and qualitative data, we constructed a joint display integrating the results of our survey and interviews thematically, and we produced an integrated narrative summary of identified themes [14,19].

### Inclusivity in global research

Additional information regarding the ethical, cultural, and scientific considerations specific to inclusivity in global research is included in the Supporting Information (S5 Text).

## Results

### Demographics

We contacted 27 FETPs and FETP networks, among which 14 (51.9%) distributed our survey among their FETP trainees and graduates. As a result, 86 respondents began the survey; out of these, three respondents (3.5%) requested that their responses be deleted before completing the survey. Among the remaining 83 responses, we removed 27 (32.5%) additional ones from our analysis, including one respondent (1.2%) who was from outside the Asia-Pacific region, one of two responses from the same respondent (1.2%), and 25 others (30.1%) that did not provide responses to the Likert scale questions or free text responses, leaving 56 responses (67.4%) for our analysis. Among these, 28 respondents (50.0%) also provided free text responses. As the survey was disseminated via FETP program directors, we were unable to determine the total number of individuals who received the invitation. Therefore, we could not calculate a formal response rate.

Most respondents were 25–44 years old (44, 78.6%) male (35, 62.5%) and had a master's or clinical degree as their highest education level (41, 73.2%) (Table 1). In addition, most respondents had completed FETP-Advanced as their highest FETP training program (33, 58.9%) and were FETP graduates (41, 73.2%) as opposed to current trainees. Ten FETP training countries were represented in the survey; the three most represented were Bangladesh (11, 19.6%), India (11, 19.6%), and the Philippines (8, 14.3%). Most respondents reported currently working in a government public health office (42, 75.0%), with the majority working at the subnational level (25, 44.6%). Nineteen countries or territories of work were represented in the survey; the three most represented were India (10, 17.9%), Bangladesh (9, 16.1%), and Cambodia (9, 16.1%). By World Bank income classification, eight persons worked in high-income countries or territories (14.3%), eight (14.3%) worked in upper-middle-income countries or territories, and 42 (75.0%) worked in lower-middle-income countries or territories. Finally, most respondents reported working in a public health role for nine years or less (37, 66.1%).

Forty-one respondents indicated interest in being interviewed and were subsequently contacted; after being contacted, 18 (43.9%) responded, confirming their interest in being interviewed. Out of these, one could not be interviewed because of connectivity issues, and one failed to present for their scheduled interview; the remaining 16 (39.0%) were successfully interviewed. Nine interviewees (56.2%) worked in a public health or clinical capacity at the state/provincial level, five (31.2%) worked at the national level, and two (12.5%) worked in academia or in a laboratory setting (Table 2). Three interviewees (18.8%) worked in South Asia, while 13 (81.3%) worked in East Asia and the Pacific. Thirteen interviewees

**Table 1. Characteristics of survey respondents (n = 56).**

| Category | Number of respondents (%) |
|---|---|
| *Age (years)* | |
| ≤24 | 0 (0.0) |
| 25-34 | 22 (39.3) |
| 35-44 | 22 (39.3) |
| 45-54 | 7 (12.5) |
| 55-64 | 1 (1.8) |
| 65+ | 0 (0.0) |
| No response | 4 (7.1) |
| *Gender* | |
| Male | 35 (62.5) |
| Female | 16 (28.6) |
| No response or prefer not to say | 5 (8.9) |
| *Highest education level* | |
| Associate or bachelor's degree (not including medical or veterinary degree) | 5 (8.9) |
| Postgraduate certificate or diploma | 4 (7.1) |
| Master's degree | 22 (39.3) |
| Clinical degree (including medical or veterinary degree) | 19 (33.9) |
| Doctorate degree (including PhD) | 1 (1.8) |
| No response or prefer not to say | 5 (8.9) |
| *Highest FETP training program participating in or completed* | |
| FETP-Frontline or Basic | 5 (8.9) |
| FETP-Intermediate | 14 (25.0) |
| FETP-Advanced | 33 (58.9) |
| FETP for Veterinarians | 1 (1.8) |
| Not listed or prefer not to say | 3 (5.4) |
| *FETP training status* | |
| Current trainee | 15 (26.8) |
| Graduate | 41 (73.2) |
| *FETP training country* | |
| Australia | 2 (3.6) |
| Bangladesh | 11 (19.6) |
| Cambodia | 7 (12.5) |
| India | 11 (19.6) |
| Mongolia | 5 (8.9) |
| Papua New Guinea | 3 (5.4) |
| Philippines | 8 (14.3) |
| Singapore | 1 (1.8) |
| Thailand | 6 (10.7) |
| United States Pacific territory | 1 (1.8) |
| No response | 1 (1.8) |
| *FETP training country by income classification[a]* | |
| High-income | 4 (7.1) |
| Upper-middle-income | 11 (19.6) |
| Lower-middle-income | 40 (71.4) |
| Low-income | 0 (0.0) |
| Unknown | 1 (1.8) |

*(Continued)*

**Table 1.** (Continued)

| Category | Number of respondents (%) |
|---|---|
| *Current work setting(s)[b]* | |
| Academic/university | 5 (8.9) |
| Government public health office | 42 (75.0) |
| Local/district office | 13 (23.2) |
| State/provincial office | 12 (21.4) |
| National health ministry/department | 17 (30.4) |
| Hospital or clinic | 6 (10.7) |
| Laboratory | 2 (3.4) |
| Non-governmental or international organization | 4 (7.1) |
| Prefer not to say | 1 (1.8) |
| *Current work country/countries[b]* | |
| American Samoa (United States territory) | 1 (1.8) |
| Australia | 2 (3.6) |
| Bangladesh | 9 (16.1) |
| Bhutan | 1 (1.8) |
| Cambodia | 9 (16.1) |
| China | 1 (1.8) |
| Federated States of Micronesia | 1 (1.8) |
| Guam (United States territory) | 1 (1.8) |
| India | 10 (17.9) |
| Japan | 1 (1.8) |
| Marshall Islands | 1 (1.8) |
| Mongolia | 5 (8.9) |
| Myanmar | 1 (1.8) |
| Palau | 1 (1.8) |
| Papua New Guinea | 3 (5.4) |
| Philippines | 8 (14.3) |
| Singapore | 1 (1.8) |
| Thailand | 1 (1.8) |
| United Kingdom | 1 (1.8) |
| No response or prefer not to say | 2 (3.6) |
| *Current work country/countries by income classification[a]* | |
| High-income | 8 (14.3) |
| Upper-middle-income | 8 (14.3) |
| Lower-middle-income | 42 (75.0) |
| Low-income | 0 (0.0) |
| Unknown | 2 (3.6) |
| *Number of years worked in public health role* | |
| Less than 5 years | 16 (28.6) |
| 5–9 years | 21 (37.5) |
| 10–14 years | 10 (17.9) |
| 15 years or more | 8 (14.3) |
| Prefer not to say | 1 (1.8) |

[a]World Bank country income level.

[b]Respondents could indicate multiple responses.

**Table 2. Characteristics of interview participants (n = 16).**

| Interviewee | Work setting | Region[a] | Country income classification[b] |
|---|---|---|---|
| 1 | Academic/laboratory | South Asia | Lower-middle |
| 2 | Public health (national) | East Asia and Pacific | Lower-middle |
| 3 | Academic/laboratory | South Asia | Lower-middle |
| 4[c] | Public health (national) | South Asia | Lower-middle |
| 5 | Public health (provincial/regional) | East Asia and Pacific | Lower-middle |
| 6 | Public health (provincial/regional) | East Asia and Pacific | Lower-middle |
| 7 | Public health (national) | East Asia and Pacific | Lower-middle |
| 8 | Public health (provincial/regional) | East Asia and Pacific | Upper-middle |
| 9 | Public health (national) | East Asia and Pacific | Upper-middle |
| 10 | Public health (provincial/regional) | East Asia and Pacific | Lower-middle |
| 11 | Public health (provincial/regional) | East Asia and Pacific | Lower-middle |
| 12 | Public health (provincial/regional) | East Asia and Pacific | Lower-middle |
| 13 | Public health (provincial/regional) | East Asia and Pacific | Lower-middle |
| 14 | Public health (national) | East Asia and Pacific | Lower-middle |
| 15 | Public health (provincial/regional) | East Asia and Pacific | Lower-middle |
| 16 | Public health (provincial/regional) | East Asia and Pacific | High |

[a]World Bank region.

[b]World Bank country income classification.

[c]Previously worked in Asia-Pacific but now working elsewhere.

(81.3%) worked in a World Bank lower-middle-income country, two (12.5%) worked in an upper-middle-income country, and one (6.3%) worked in a high-income country.

We identified key factors from the survey for further investigation in the interviews according to impact scores and free text responses that repeatedly highlighted important barriers and enablers. The results of the survey indicated that key factors impacting reporting were the availability of sufficient resources and personnel as well as respondents feeling disincentivized from reporting (including because of concerns over reputational or economic damage). On the other hand, the results highlighted increased capacity (such as increased personnel and resources) as key reporting enablers. We subsequently used these findings to inform development of the semi-structured interview guide, which included questions exploring these factors specifically as well as general questions to elicit additional barriers and enablers known or experienced by each interviewer.

## Thematic barriers and enablers

Based on the quantitative survey findings and merged qualitative data, we elicited four thematic barriers – lack of capacity, behavioural barriers, political and socioeconomic barriers, and bureaucratic barriers – and three thematic enablers – building capacity, building a culture of reporting, and political and bureaucratic support. We provide joint displays of quantitative and qualitative findings for the thematic barriers and enablers in Tables 3 and 4, respectively, and we provide Likert scale responses disaggregated by country of work income level in the Supporting Information (S1–S3 Tables).

## Lack of capacity

Respondents highlighted capacity limitations as a key barrier to outbreak reporting. In the survey, insufficient staff to report was the barrier most described as having a high impact on outbreak notifications (28, 50.0%), with half or more of

**Table 3. Joint display of survey results and illustrative quotes on barriers to outbreak reporting, arranged by theme.**

| Category | Number of respondents (n = 56) | | | | Illustrative qualitative responses[a] |
|---|---|---|---|---|---|
| | High impact (%) | Some impact (%) | No impact (%) | Unsure or no answer (%) | |
| _Lack of capacity_ | | | | | |
| Staff too busy to report an outbreak | 23 (41.1) | 26 (46.4) | 4 (7.1) | 3 (5.4) | _[P]rimary level healthcare workers are occupied with several responsibilities including providing healthcare service for a large population. So, it is not unusual that event that require reporting missed or ignored by staffs, particularly those do not fall under any active surveillance. [Survey free text response]_ |
| Not enough available staff to report an outbreak | 28 (50.0) | 18 (32.1) | 7 (12.5) | 3 (5.4) | _[T]here are areas…especially in the low-income municipalities in my country that has a limited number of personnel…to do outbreak response and monitoring. And what really happens is that, for example…I'll be handling the…disease surveillance monitoring. But I'm also handling the, the vaccination program, the rabies program, and other programs because most of the time people are handling…three to five or sometimes six programs, and it's all by themselves. [Interviewee 6]_ |
| Reporting is too complicated, difficult, or time-consuming | 20 (35.7) | 30 (53.6) | 4 (7.1) | 2 (3.6) | _Medical officers are afraid and don't want to go through the burden of documentation. [Survey free text response]_ |
| Staff do not know what requires reporting | 22 (39.3) | 26 (46.4) | 3 (5.4) | 5 (8.9) | _When I [was] first exposed in the field, our team member in the field level, they did not know how to report, or they didn't understand which type of event that they can report to provincial level or upper level. [Interviewee 13]_ |
| Staff do not know how to report an outbreak | 24 (42.9) | 25 (44.6) | 4 (7.1) | 3 (5.4) | |
| Staff do not know to whom to report an outbreak | 21 (37.5) | 20 (35.7) | 14 (25.0) | 1 (1.8) | |
| Lack of surveillance resources to detect an outbreak | 27 (48.2) | 22 (39.3) | 6 (10.7) | 1 (1.8) | |
| Lack of laboratory resources to identify outbreak pathogen | 27 (48.2) | 19 (33.9) | 8 (14.3) | 2 (3.6) | _Diagnostic facilities at sub-district level are not up to the mark. Therefore, when a physician suspect something worth reporting, he or she cannot confirm it due to lack of tools. [Survey free text response]_ |
| Lack of resources to report an outbreak (for example: access to telephone, computer, internet, appropriate forms) | 17 (30.4) | 26 (46.4) | 12 (21.4) | 1 (1.8) | _Areas with no…Internet or computer would be having difficulty to report…cases, and usually there is a delay of about a week or two before we can get the report cause the personnel is going in an area or traveling to the mainland because we have a lot of island municipalities… [Interviewee 6]_ |
| Lack of funding for outbreak reporting[b] | – | – | – | – | _Now that the pandemic's over, we don't have funding anymore, that in most places it kind of went back the way it was before. [Interviewee 12]_ |
| _Behavioural barriers_ | | | | | |
| Staff not motivated to report outbreaks | 21 (37.5) | 27 (48.2) | 7 (12.5) | 1 (1.8) | _[S]urveillance is something that is not reimbursed by health insurance, which means there is no incentive for health facilities or health workers to report. [Survey free text response]_ |
| Staff afraid of being punished for reporting outbreaks or being blamed for the outbreak | 24 (42.9) | 19 (33.9) | 11 (19.6) | 2 (3.6) | _In low- and middle-income countries (LMICs), the staff involved in disease control tends to conceal outbreaks. They believe that an outbreak is a reflection of their own mistake, which could negatively impact their job responsibilities and achievements. [Survey free text response]_ |
| Staff pressured to not report outbreaks | 22 (39.3) | 17 (30.4) | 13 (23.2) | 4 (7.1) | _For…example, if the hospital [report to the province], but the province did not want to report to the region, they can block somehow. [During an] outbreak of chikungunya…which is…not a usual event…they report to the province. But the province…I think they are denied [permission] to report…further into the central level because they think…that it should not happen, because if the new, this kind of new disease happened in the province…the head of the provincial hospital should have a problem." [Interviewee 8]_ |

_(Continued)_

| | Number of respondents (n = 56) | | | | Illustrative qualitative responses[a] |
|---|---|---|---|---|---|
| Staff prefer responding to outbreak before reporting[b] | – | – | – | – | [If] we can do our disease control for ourselves, we tend to…implement that disease control before reporting…And the provincial health office, they act as a filter to determine whether they respond to that particular outbreak for themselves or even further notification to the…regional office or even the Ministry of Health. [Interviewee 8] We can clarify, we can confirm, we can respond, and then we do the…report letter to…WHO by [one] week or…two week. [Interviewee 13] |
| Community members do not approach traditional health workers when ill[b] | – | – | – | – | [T]here is a little bit of a tendency to underreport…from the most…basic health worker who is in the village…[T]he health seeking behaviour of the public is such, of the community is such, that they hit the health system a little late or…when things have already gone a little out of control because…the conventional convention…is to follow the traditional system of medicine or the…local available practitioners who may not be the actual allopaths who would go or be a part of the reporting system." [Interviewee 3] |
| Political and socioeconomic barriers | | | | | |
| Government not interested in encouraging outbreak reporting or making outbreak reporting easier | 23 (41.1) | 18 (32.1) | 11 (19.6) | 4 (7.1) | [T]he decision to report outbreaks often depends on the organizational leadership. In many cases, the relevant authorities are reluctant to report outbreaks due to a lack of high-level political motivation. [Survey free text response] |
| Fear of economic damages from reporting (for example: losses to trade or tourism) | 20 (35.7) | 24 (42.9) | 10 (17.9) | 2 (3.6) | I did investigate an outbreak before, a cholera outbreak, in a municipality where they didn't want to declare officially it was an outbreak because that municipality was…a tourist destination... And they were afraid that it might affect tourism and local economy." [Interviewee 12] |
| Fear of media exposure following reporting | 23 (41.1) | 24 (42.9) | 7 (12.5) | 2 (3.6) | Everyone tends to avoid media limelight for the wrong reasons; hence some underreporting is there. [Survey free text response] |
| Fear of negative political ramifications from reporting[b] | – | – | – | – | The most difficult part in [declaring] an outbreak is that your head of office [or] politicians would not want to [declare] it as it would affect their "credibility". [Survey free text response] |
| Bureaucratic barriers | | | | | |
| Difficulty coordinating with other agencies, ministries, or sectors | 21 (37.5) | 26 (46.4) | 6 (10.7) | 3 (5.4) | In relation to zoonotic diseases, there is a lack of coordination and collaboration among sectors, and the systematic sharing of information is still lacking. [Survey free text response] |
| Staff lack authority to report outbreaks | 20 (35.7) | 22 (39.3) | 11 (19.6) | 3 (5.4) | [I]t seems to me that in order to write an outbreak report following an outbreak response, we must obtain authorization from our higher authority. For this reason, staff are becoming busy with their other work and have also lost interest in reporting, and they also have apprehension. [Survey free text response] |
| Lack of reporting mandate, regulations, or legislation | 23 (41.1) | 17 (30.4) | 12 (21.4) | 4 (7.1) | [S]ometimes it's [limited] by law or by regulation that…just certain people have authority to access the…reporting program. [Survey free text response] |
| Concerns about protecting patient privacy | 8 (14.3) | 29 (51.8) | 16 (28.6) | 3 (5.4) | Because of the data privacy law…public health officials cannot get information about outbreaks, especially for doing contact tracing. [Interviewee 6] |

[a]Filler and irrelevant words removed, clarifying words inserted, and identifying information redacted.

[b]Not asked in survey.

**Table 4. Joint display of survey results and illustrative quotes on enablers of outbreak reporting, arranged by theme.**

| Category | Number of respondents (n=56) | | | | Illustrative qualitative responses[a] |
|---|---|---|---|---|---|
| | High impact (%) | Some impact (%) | No impact (%) | Unsure or no answer (%) | |
| _Building capacity_ | | | | | |
| Easy ways to report outbreaks (for example, simplified or electronic reporting) | 36 (64.3) | 10 (7.9) | 3 (5.4) | 7 (12.5) | _[I]n my earlier years in my career, there used to be a lot of problems when remote areas had to…send in their report…had a lot of delays. But not anymore. Everybody is connected, and it's all very instantaneous. [Interviewee 3]_ |
| Designated persons(s) responsible for reporting an outbreak | 37 (66.1) | 10 (17.9) | 2 (3.6) | 7 (12.5) | _[A]ll the health [facilities] have trained surveillance focal person[s], so they are mandated to [report] all the events. [Survey free text response]_ |
| Specific training about what to report and how to report an outbreak | 40 (71.4) | 9 (16.1) | 2 (3.6) | 5 (8.9) | _We sensitize them on...diseases of concerns, emerging and reemerging...and we train them to monitor them...every week. [Interviewee 15]_ |
| Sufficient surveillance resources to detect an outbreak | 40 (71.4) | 9 (16.1) | 3 (5.4) | 4 (7.1) | _We have resources to detect outbreak and Field Epis in all provinces. [Survey free text response]_ |
| Sufficient laboratory resources to identify outbreak pathogen | 33 (58.9) | 16 (28.6) | 3 (5.4) | 4 (7.1) | |
| Appropriate funding for outbreak reporting[b] | - | - | - | - | _During COVID-19…our health ministry had a lot more funding because of the pandemic, so we were able to hire staff paid by national government but deployed to all these municipalities and cities. And they acted as the surveillance officers there. So reporting improved at that time. [Interviewee 12]_ |
| Use of non-official sources of information[b] | - | - | - | - | _So there are also occasions where we don't get official reports through either of our indicator event base, but we capture it first through a news report coming from a local media, and that's where we start the verification and response. [Interviewee 12]_ |
| _Building a culture of reporting_ | | | | | |
| Instruction on the importance of reporting an outbreak | 35 (62.5) | 11 (19.6) | 3 (5.4) | 7 (12.5) | _What makes reporting…easier is…continuous sensitization or educating [inaudible] those at the facility as well as sensitizing the communities, especially in the events of an…unusual cause of deaths or unusual diseases that are [affecting] the community. [Interviewee 15]_ |
| Encouragement to report from more senior official(s) | 33 (58.9) | 14 (25.0) | 3 (5.4) | 6 (10.7) | _I always motivate my [inaudible] that I cover in my province, I always ask them to report immediately and early about what happened. Do not hesitate to report because preliminary report is not right or wrong. We just want to know what happened in the community. [Interviewee 13]_ |
| Feedback on report quality (including information on what was reported well and how reports can be improved) | 29 (51.8) | 16 (28.6) | 4 (7.1) | 7 (12.5) | |
| Feedback on outbreaks reported (including epidemiological and outbreak response information) | 34 (60.7) | 13 (23.2) | 4 (7.1) | 5 (8.9) | _But now with the field epi training program that we facilitating at the…field level, and also we are communicating with…workers on the ground that whatever reports comes from, either from a community or from someone on the street…we will always get that report back to you to verify. [Interviewee 10]_ |

_(Continued)_

**Table 4.** (Continued)

| Category | Number of respondents (n=56) | | | | Illustrative qualitative responses[a] |
|---|---|---|---|---|---|
| | High impact (%) | Some impact (%) | No impact (%) | Unsure or no answer (%) | |
| Sufficient authority to report an outbreak | 30 (53.6) | 16 (28.6) | 3 (5.4) | 7 (12.5) | *Sometime if I think that disease is too dangerous to be…reported on the formal system, we bypass the [inaudible]…So they deny to report and they tend to [say]…the reporting doctor is not so competent. They might…have a mistake in the report. Let's…think about the dengue or another common disease…[A]nd it happened until the…doctor reporting to the Ministry of Health…because they know someone. And the…national level got to investigate. [Interviewee 8]* |
| Reimbursing or rewarding persons who report outbreaks | 25 (44.6) | 13 (23.2) | 9 (16.1) | 9 (16.1) | *The other is literally recognizing people who have reported, giving them recognition, giving them some incentive or definitely recognition so that it encourages others to follow suit. It's not always done. It's not always done. And many things are taken for granted. [Interviewee 3]* |
| Punishing persons who fail to report outbreaks | 11 (19.6) | 22 (39.3) | 13 (23.2) | 10 (17.9) | |
| Providing political and bureaucratic support | | | | | |
| Presence of reporting mandate, regulations, or legislation | 30 (53.6) | 17 (30.4) | 3 (5.4) | 6 *(10.7)* | *[H]ealth facilities are required…to report notifiable diseases through our…surveillance system, and…this includes both government and private…national and locally run…government facilities. So…this goes down to health centre levels; so down at the village level…everyone is required to report…notifiable diseases by law. [Interviewee 12]*<br>*In line with…several types of governmental order and regulation, from primary level, it should be reported to next level…if it's zoonotic disease, it has to be reported to human health and animal health sector. [Interviewee 9]* |
| Government interest in encouraging outbreak reporting or making outbreak reporting easier | 37 (66.1) | 12 (21.4) | 2 (3.6) | 5 (8.9) | *Unless the highest authority gives an order…the local authorities cannot implement it as an initiative or an innovation…So if you want anything to change or anything to be new to be implemented, it has to come from the ministries. [Interviewee 3]* |
| Good coordination between other agencies, ministries, or sectors | 38 (67.9) | 9 (16.1) | 2 (3.6) | 7 (12.5) | *So we are training our local officers that they not only keep watch on health matters, but they also keep watch on…any unusual events on environment, animals, plants. Whatever that's happening in the community, we have those trained personnels on the ground. And we asked them to keep out an eye on what's happening and then report through their respective channels like…the agriculture people, plant health people, they report to the network, and then obviously it will reach us. And we send this information if it deals with human beings or…for [inaudible] health matters, we…report to our network. [Interviewee 10]*<br>*[T]he data harmonisation is really important. Like [redacted] and [redacted] had the same base data platform, and that was incredibly useful for us. We could learn from each other, we had regular meetings talking about…how did you guys do this and…what will you be doing with this? And we would share lessons all the time, and that should really have been happening on a national basis… [Interviewee 16]* |

*(Continued)*

PLOS Global Public Health

**Table 4.** (Continued)

| Category | Number of respondents (n=56) | | | | Illustrative qualitative responsesª |
|---|---|---|---|---|---|
| | High impact (%) | Some impact (%) | No impact (%) | Unsure or no answer (%) | |
| Actions to protect patient privacy (for example: promising not to share patient data or promising to destroy patient data after period of time) | 25 (44.6) | 18 (32.1) | 6 (10.7) | 7 (12.5) | |

ªFiller and irrelevant words removed, clarifying words inserted, and identifying information redacted.

ᵇNot asked in survey.

the respondents working in lower-middle- and upper-middle-income countries noting this as having a high impact; staff not knowing how to report (24, 42.9%), being too busy to report (23, 41.1%), and not knowing what to report (22, 39.3%) were also notable barriers with a high impact. The qualitative responses further elaborated on these findings, noting that a major capacity barrier relates to personnel with notification responsibilities, particularly knowledge lapses regarding outbreak reporting, including not knowing how and what to report or that one is even required to report. In addition, these responses discussed too few or overburdened personnel, as well as high personnel turnover, as another personnel barrier, noting that reporting was particularly challenging for health care workers at the ground level, who must juggle notification responsibilities with clinical care work.

Another capacity barrier highlighted by survey respondents as having a high impact on reporting included lack of infrastructure needed for generating reports, including lack of surveillance (27, 48.2%) and laboratory (27, 48.2%) resources, with half of upper-middle- and high-income respondents noting that shortfalls with the latter had a high impact on outbreak notifications. Indeed, the qualitative responses noted that lack of laboratory capacity can delay pathogen diagnosis during a new outbreak, slowing response planning. Seventeen survey respondents (30.4%) reported that the lack of resources for physically reporting an outbreak had a high impact on reporting. This was further illustrated in the qualitative data, which highlighted the unavailability of reporting forms and inadequate communication and transportation infrastructure. While the qualitative responses from persons from lower-middle-income countries noted that lack of utilities such as internet connectivity or electricity can slow reporting of new events from more remote areas, those from upper-middle-income and high-income countries noted challenges with having appropriate software or electronic notification systems in place. A qualitative observation not noted in the survey questions was that challenges in obtaining appropriate funding made it more difficult to build the required capacity for outbreak notifications.

### Behavioural barriers

Another set of barriers was behavioural. In the survey, 21 respondents (37.5%) indicated that lack of motivation to report had a high impact on outbreak reporting. The qualitative responses elaborated further, noting that this challenge may stem from not only a lack of intrinsic motivation or incentive to report, such as financial incentives, but also a taboo against reporting, particularly for lower-level officials who might fear retaliation for calling attention to the outbreak or being blamed for the outbreak itself (i.e., for failing to prevent the outbreak). Indeed, 24 survey respondents (42.9%) agreed that fear of being punished or blamed for reporting had a high impact on outbreak notifications, while 22 respondents (39.3%) noted that pressure to not report is also a barrier with a high impact. Additional qualitative insights revealed that local officials might choose to defer reporting if they believe they can handle control efforts on their own. Similarly, national governments might choose to delay notifications to WHO until after they have begun their outbreak response, indicating that certain governments view international outbreak notifications as a lower priority. The qualitative responses further indicated that at

the local level, community members might choose to not notify local officials because those members might not appreciate the need to report or instead rely on traditional practices; this can prevent health care workers from learning of outbreak cases until well after the outbreak has begun.

## Political and socioeconomic barriers

Twenty survey respondents (35.7%) indicated that fear of economic damage had a high impact on outbreak reporting. The qualitative responses elaborated further, noting that fear of economic damage, such as loss of tourism, can motivate local governments to prevent the upward flow of outbreak notifications, particularly where local governments have more power to control reporting, such as in countries with decentralized or federalized political structures. Of note, while most survey respondents working in lower-middle- and upper-middle-income countries noted that fear of economic damages had at least some impact on reporting, most working in high-income countries felt that economic fears had no impact. In addition, 23 survey respondents (41.1%) noted that fear of media exposure had a high impact on outbreak notifications. Indeed, the qualitative findings revealed that politicians feared reputational loss because of media, public, or official scrutiny following reports of an outbreak. They further noted that local reporting officials might delay transmission of outbreak information to higher levels, or governments might withhold disease information from the public during an outbreak investigation. Also, 23 survey respondents (41.1%) felt that government disinterest in outbreak reporting had a high impact on outbreak notifications; similarly, the qualitative responses noted that the failure of government officials to understand the importance of outbreak reporting, and disease surveillance more largely, can undermine outbreak notifications.

## Bureaucratic barriers

Twenty-one survey respondents (37.5%) noted that difficulty coordinating with other agencies, ministries, or sectors had a high impact on outbreak reporting, and 20 (35.7%) noted that staff lacking authority to report outbreaks had a high impact. Several qualitative responses identified various bureaucratic barriers that make reporting more difficult either vertically (from the local to the national level) or horizontally (between sectors or agencies). They noted that strict reporting hierarchies can make it challenging to report events directly from the local to the national level, potentially delaying outbreak responses, while coordination barriers can inhibit the sharing of data across the human, animal, and environmental sectors, such as during zoonotic outbreaks. Twenty-three survey respondents (41.1%) also noted that lack of a reporting mandate, regulations, or legislation had a high impact on outbreak notifications. One qualitative response specifically noted that the absence of explicit rules or legislation to facilitate data harmonisation and sharing can inhibit cooperation between different reporting jurisdictions. On the other hand, some qualitative responses noted that the perceived need to abide by data privacy laws can stymie the sharing of crucial patient data during outbreaks, although patient privacy concerns were the barrier least described as having a high impact on outbreak notifications in the survey (8, 14.3%).

## Building capacity

Across both the quantitative and qualitative findings, addressing capacity barriers was one of the most important ways to facilitate outbreak reporting. Forty survey respondents (71.4%) noted that training on outbreak reporting has a high impact in facilitating outbreak notifications, the highest among individual or team-level enablers; this was reinforced by qualitative accounts emphasising the importance of training in facilitating outbreak reporting. Several survey respondents also indicated that appropriate reporting infrastructure, including adequate surveillance resources (40, 71.4%) and laboratory resources (33, 58.9%) as well as easier ways to report (36, 64.3%) had a high impact in facilitating outbreak notifications. The qualitative responses further elaborated on this, noting the importance of having personnel with surveillance responsibilities, including epidemiologists and health care personnel, as well as adequate telecommunications capacity to support detection and reporting. Thirty-seven survey respondents (66.1%) also indicated the importance of having designated reporting personnel, which was substantiated in the qualitative findings. However, the qualitative responses

also highlighted the importance of adequate financing as well as the ability to scan media reports for new outbreaks to help officials detect outbreaks before they are officially reported through normal notification channels. These insights were unique to the qualitative phase and were not investigated in the survey.

### Building a culture of reporting

Both the quantitative and qualitative findings highlighted the importance of fostering a culture of reporting among those responsible for outbreak notifications as a key factor in strengthening the reporting process. Thirty-five respondents (62.5%) noted that instruction on the importance of outbreak reporting had a high impact on facilitating notifications, while 33 (58.9%) stated that encouragement to report had a high impact. Additionally, 34 (60.7%) indicated that providing updates to reporting officials on outbreaks they reported had a high impact, while 29 (51.8%) noted that feedback on report quality also had a high impact. These findings were reinforced by the qualitative responses, which emphasized the importance of building norms or expectations around reporting; they specifically noted that encouraging personnel to report through acknowledgement of work performed, including appropriate feedback, can facilitate outbreak notifications. Thirty (53.6%) survey respondents indicated that sufficient authority to report an outbreak was an enabler with a high impact. Relatedly, the qualitative responses noted that a well-established norm around outbreak notifications can encourage persons to report outside official channels, including where intermediary officials in the notification chain block reporting to the national level. Of note, while 25 (44.6%) survey respondents indicated that reimbursing or rewarding reporting had a high impact on facilitating outbreak notifications, few respondents noted that punishing failures to report had a high impact (11, 19.6%). Similarly, the qualitative findings largely supported the importance of providing positive incentives for reporting but did not endorse the threat of punishment as an incentive.

### Political and bureaucratic support

Finally, both the quantitative and qualitative findings described the importance of providing political and bureaucratic support for outbreak reporting. Survey respondents noted government interest in reporting (37, 66.1%) and reporting mandates, regulations, or legislation (30, 53.6%) had a high impact in facilitating outbreak notifications; indeed, all the survey respondents working in high-income countries noted that appropriate governance had a high impact on reporting. The qualitative responses supported this, highlighting the importance of not only mandatory vertical reporting but also horizontal information sharing across sectors. While 38 survey respondents (67.9%) noted the importance of good coordination between agencies, ministries, or sectors as having a high impact in facilitating outbreak notifications, the qualitative responses specifically noted the importance of training local officials to take an integrated One Health approach to surveillance and reporting to encourage information sharing across the human, animal, and environmental sectors. Finally, while 25 survey respondents (44.6%) noted that actions to protect patient privacy had a high impact on reporting, this theme did not emerge in the qualitative findings.

### Discussion

This is the first study we are aware of that explored the unique knowledge, perspectives, and experiences of field epidemiologists across the Asia-Pacific region on the technical and non-technical barriers and enablers of outbreak reporting. We found several barriers to effective outbreak reporting at the subnational level, including capacity, behavioural, political, socioeconomic, and bureaucratic barriers. Respondents also identified key reporting enablers, many of which related to building capacity but also to building a culture of reporting and ensuring political and bureaucratic support.

We found that capacity barriers continue to be substantial impediments to outbreak reporting. As with previous studies, we found that insufficient personnel, lack of diagnostic capacity, logistical challenges, and lack of provider knowledge and time were major barriers to timely information sharing at the subnational level [7,20–23]. An additional study found similar

concerns among the National Focal Points (NFPs) responsible for reporting significant public health events to WHO in accordance with the IHR [24]. Challenges in obtaining appropriate funding made it more difficult to build outbreak reporting capacity, which is also well-described in the literature [25,26]. Our findings thus demonstrate that further investments to improve these capacities remain necessary.

We additionally identified challenges to the vertical communication of outbreak reports from the subnational to the national level as important barriers, which also corroborates previous findings. These challenges included difficulties in data sharing between the subnational and national levels, devolution of public health responsibilities to the subnational level, and the requirement for approval at the subnational level to transmit information to the national level [27–29]. In addition, we identified challenges in horizontal information sharing between sectors, including the human and animal health sectors, because of organisational siloing and mistrust [5,24,27,30–33]. One area of dissonance between the quantitative and qualitative findings was the perceived need to respect patient privacy, which although supported in the qualitative findings was largely downplayed in the survey findings. This dissonance likely flows from the wording of the survey question, which emphasized respecting patient privacy, while the qualitative responses emphasized data privacy laws as a barrier to information sharing, in line with previous studies [5,34,35]. These findings highlight the need for appropriate legislation, rules, and policies to facilitate timely vertical and horizontal outbreak data sharing.

Behavioural barriers also emerged as significant factors affecting outbreak reporting in our study. These included lack of encouragement or active punishment for outbreak reporting, lack of motivation to report, and fear of blame or punishment for reporting an outbreak, which others have previously identified [21,36,37]. One significant, and potentially novel, finding was that at the national level, government officials have delayed outbreak notification until after their outbreak response is underway, despite the IHR requiring notification to WHO within 24 hours of an IHR NFP being made aware of a potential public health event of international concern (PHEIC) [1]. This finding suggests that some governments might hesitate to acknowledge an outbreak until they can convey that it is under control, perhaps to avoid trade or travel restrictions. Indeed, a previous study found that Southeast Asian states were hesitant to report outbreaks to WHO headquarters for fear of creating significant bureaucratic burdens and public panic that could result in economic uncertainty [38]. Additional research should investigate where and in what contexts this practice operates and whether governments actively suppress unofficial reports (e.g., media reports) until the outbreak is controlled.

The fear of economic repercussions as a barrier to reporting outbreaks at the national level is also well-described [5,24,31,38]. Our findings build on this evidence by demonstrating that this fear also impedes reporting at the subnational level, particularly among lower income countries. Previous studies have focused on how such fears impact the notification of animal diseases, including zoonoses, through concealment of outbreaks and encouragement of animal health officers to not formally report animal diseases [39,40]. We believe this is the first study to clearly describe the fear of economic repercussions as a barrier to local reporting in the human health sector. Further research should explore the presence of such barriers among localities at high risk of outbreaks, including the contexts in which these barriers emerge.

Our study identified several factors that facilitate outbreak reporting, in line with previous studies. These include improving capacity, including adequate infrastructure and well-trained personnel, willingness to report, and appropriate governance [5,21,23,41]. Financing emerged as a key enabler, with participants highlighting how limited resources constrain reporting capacity. Additionally, the ability to scan media reports was identified as an important non-official source of outbreak information [5,25,39,42]. Although our scoping review had identified this as a reporting enabler, we chose not to include this in the survey because of our decision to focus on official reporting practices in this study [6]. Nonetheless, given the repeatedly cited use of this practice among our study participants, it was important to note this in our results. Furthermore, we found that motivating persons to report through education, reminders, and feedback are vital for building a culture of reporting [21,24]. Notably, our study findings rejected punitive approaches to reporting; this contrasts with earlier studies suggesting that fear of punishment can incentivize reporting among certain groups [43,44]. Further studies should investigate under what circumstances fear of punishment would be an appropriate reporting facilitator. Our study

also highlighted the importance of sustained, structured, and appropriately resourced intersectoral cooperation as a key enabler of outbreak notifications, including between the human and animal health sectors [24,27,31]. Lastly, we found that high-level political support is critical for sustaining surveillance and reporting activities [38,45,46]. The impact of sensitizing subnational governmental officials to the importance of these activities is an important area for future research, given their outsized role in controlling the flow of outbreak information.

Our use of a mixed methods study methodology revealed insights that would unlikely have been revealed using only a quantitative or qualitative method. By first collecting quantitative data, we were able to identify the most frequently reported barriers and enablers, allowing us to design an interview guide that explored real-world experiences regarding commonly reported barriers and enablers. The use of open-ended questions in the survey also enabled us to collect qualitative insights from a larger number of participants, enriching the qualitative data from interviews. Not having the quantitative data would have likely led to more unstructured and unfocused interviews that would have risked missing key insights. On the other hand, exclusive use of a quantitative method would have reduced our ability to explore how putative barriers and enablers function in a real-world setting as well as to identify additional factors that were not captured using the Likert-scale questions.

## Limitations

This study had several limitations. First, the study sample size was too small to make detailed statistical comparisons among the different Likert scale responses based on the varying geographic, political, and socioeconomic contexts in which the respondents worked. However, the surveys were primarily designed to elicit thematic information in conjunction with the qualitative findings rather than primarily be used to make statistical inferences. To this end, we were able to use the Likert scale responses to obtain a broad perspective on various outbreak reporting barriers and enablers, which we used to guide the interviews and obtain clarifying information that enriched our overall understanding of how these factors operate in the Asia-Pacific region. Next, the study drew exclusively from the viewpoints and experiences of a convenience sample of field epidemiologists. As a result, the responses were potentially skewed towards barriers and enablers experienced by persons who are but a subset of a wider group of officials with outbreak reporting responsibilities. In addition, although an exact number of FETP trainees and graduates within our sampling frame was not available, we estimate that the total number is likely in the thousands, suggesting that we only captured a fraction of our targeted sample [8]. Nevertheless, our findings were consistent with the existing literature, underscoring the salience of the information provided by the participants. Furthermore, although they only represent a portion of reporting officials, field epidemiologists who report outbreaks today will likely receive those reports and direct subsequent responses as they rise in seniority [47]; as such, they represent a critical group for not only understanding the barriers affecting outbreak reporting at the ground level but also for addressing those barriers as their careers progress. An additional limitation was that the roles of outbreak reporting officials might differ between countries, affecting the comparability of responses; however, the interviewees often echoed similar experiences, indicating that their roles and responsibilities were likely comparable. Our findings could also have been biased by overrepresentation by one or more countries in this study, and there might have been a selection bias in favour of persons who had the means and motivation to discuss outbreak reporting barriers and enablers. However, we reached saturation in our interviews, which is a sign that although the sample size was small and there may have been overrepresentation from some countries, we were able to elicit common themes that likely extend across different reporting environments. Another limitation was that our use of a nested sampling design involving interviews with survey respondents excluded interviews of those who did not participate in the survey, who might have had divergent views from those who participated. We also conducted our study in English, which might have excluded important insights among field epidemiologists with limited to no English fluency. Finally, social desirability bias might have led interviewees to refrain from discussing sensitive topics; however, the interviewees repeatedly highlighted non-technical barriers of a potentially sensitive nature, indicating their comfort in discussing sensitivities around reporting.

## Conclusions

We examined outbreak reporting barriers and enablers at the subnational level across the Asia-Pacific region. While our findings identified previously reported outbreak reporting barriers and enablers, we also identified novel barriers at the subnational level, including the influence of economic and reputational considerations when making the decision to report, particularly among lower income countries. We also found evidence that national governments weigh considerations other than their notification obligations under the IHR when deciding whether to report an outbreak. This is the first study that we are aware of that examined the full scope of outbreak reporting barriers and enablers in the Asia-Pacific region and lays the foundation for more in-depth studies. Future studies should examine other regions as well as why and how subnational governments inhibit reporting, how the national level influences the decision to report at the subnational level, and how to inculcate a culture of reporting among governmental and reporting officials at all levels.

## Supporting information

**S1 Text. Study protocol.**
(PDF)

**S2 Text. Survey on barriers and enablers of outbreak reporting.**
(PDF)

**S3 Text. Guide for interviews on barriers and enablers of outbreak reporting.**
(PDF)

**S4 Text. Coding scheme for interviews on barriers and enablers of outbreak reporting.**
(PDF)

**S5 Text. Inclusivity in global research questionnaire.**
(PDF)

**S1 Table. Survey responses for barriers and enablers to outbreak reporting among respondents working in lower-middle-income countries (n = 42).**
(PDF)

**S2 Table. Survey responses for barriers and enablers to outbreak reporting among respondents working in upper-middle-income countries (n = 8).**
(PDF)

**S3 Table. Survey responses for barriers and enablers to outbreak reporting among respondents working in high-income countries (n = 6).**
(PDF)

**S1 File. GRAMMS Checklist.** Good Reporting of a Mixed Methods Study (GRAMMS) checklist.
(PDF)

## Acknowledgments

We thank Amy Parry and Chia-Ping Su for helping develop the survey instrument. This research was previously presented in part at the 2024 Global Health Security Conference in Sydney, Australia (18 June – 21 June 2024).

## Author contributions

**Conceptualization:** Amish Talwar.

**Data curation:** Amish Talwar.

**Formal analysis:** Amish Talwar, Matthew M. Griffith.

**Investigation:** Amish Talwar.

**Methodology:** Amish Talwar.

**Project administration:** Amish Talwar.

**Supervision:** Rebecca Katz, Martyn D. Kirk, Tambri Housen.

**Writing – original draft:** Amish Talwar.

**Writing – review & editing:** Matthew M. Griffith, Rebecca Katz, Martyn D. Kirk, Tambri Housen.

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
