## [Decision Letter · Decision Letter 0]

25 Nov 2025

PGPH-D-25-03067

The barriers and enablers of outbreak reporting in the Asia-Pacific region: a mixed methods study of field epidemiologists

Dear Dr. Talwar,

Thank you for submitting this very interesting and well-written manuscript to PLOS Global Public Health. After careful consideration, we feel that it has merit and meets PLOS Global Public Health’s publication criteria subject to addressing a few minor reviewer comments indicated below. Therefore, we invite you to submit a revised version of the manuscript that addresses the below points raised during the review process.

A letter that responds to substantive points raised by the reviewers. You should upload this letter as a separate file labeled 'Response to Reviewers'.A marked-up copy of your manuscript that highlights changes made to the original version. You should upload this as a separate file labeled 'Revised Manuscript with Track Changes'.An unmarked version of your revised paper without tracked changes. You should upload this as a separate file labeled 'Manuscript'.

We look forward to receiving your revised manuscript.

Kind regards,

Roojin Habibi, Ph.D., M.Sc., J.D.

Academic Editor

Journal Requirements:

2. Please ensure that your Ethics Statement is available in its entirety at the beginning of your Methods section, under a subheading 'Ethics Statement'.

Reviewers' comments:

Reviewer's Responses to Questions

**Comments to the Author**

1. Does this manuscript meet PLOS Global Public Health’s publication criteria?

Reviewer #1: Yes

Reviewer #2: Yes

2. Has the statistical analysis been performed appropriately and rigorously?

Reviewer #1: N/A

Reviewer #2: N/A

3. Have the authors made all data underlying the findings in their manuscript fully available (please refer to the Data Availability Statement at the start of the manuscript PDF file)?

Reviewer #1: Yes

Reviewer #2: Yes

4. Is the manuscript presented in an intelligible fashion and written in standard English?

Reviewer #1: Yes

Reviewer #2: Yes

Reviewer #1: This is an informative study on the non-technical conditions that inform reporting behaviour considerations amongst those who have the responsibility, in their national public health sectors, to detect and report suspected outbreaks.

The findings are insightful and build on/compliment existing research into the political, social and economic conditions that determine reporting behaviour.

If I had to suggest two comments to show that I read the paper (!), it would be on the conclusions that can be drawn from the sample. First, the survey was completed by those with less than 9 years of experience with a sample age quite youthful. One could suggest that reporting decisions tend to be in the remit of those with more seniority. So the vertical communication, behavioural, capacity issues and political concerns could be attributed to those with less experience and responsibility in the role, i.e. knowledge and experience will grow over time. However, a counter argument could be that the fact the 'first detectors' are identifying these limitations is a huge concern - they are the early warning mechanism. A second comment is that the sample has more provincial versus national located staff, and this could also influence the responses and findings, i.e. reporting roles and responsibility differs per country depending on their centralised/decentralised funding and regulations, which may affect the perception of non-technical barriers. However, as I said above, these are minor comments on an excellent and informative paper that I will be citing when it is hopefully published.

Reviewer #2: 1. The manuscript addresses a relevant public health question, is well structured, and presents its arguments clearly. It aligns with the journal’s scope and contributes meaningfully to the literature.

2. The manuscript appears technically sound. While my training is in law rather than qualitative or quantitative methodologies, the analytical approach is coherent and the reasoning is well developed. The conclusions follow logically from the data and analysis provided.

3. The research is presented as methodologically and ethically rigorous, and the authors draw conclusions that are consistent with the evidence they provide.

4. They have not shared their data. Their data statement says that they are maintaining the data for privacy reasons. Their statement says: "To protect participants from inadvertent identification and linkage with sensitive information elicited through this study, all data arising from this study will be retained at

the Australian National University for at least five years following publications arising

from the research. After this storage period, participant data will be de-identified and

archived at the Australian Data Archive (www.ada.edu.au) for use by other

researchers."

5. The manuscript is clearly written and well organised.

**Do you want your identity to be public for this peer review?** For information about this choice, including consent withdrawal, please see our Privacy Policy

Reviewer #1: **Yes: ** Sara E Davies

Reviewer #2: No

---

## [Editor Report · Decision Letter 1]

15 Dec 2025

The barriers and enablers of outbreak reporting in the Asia-Pacific region: a mixed methods study of field epidemiologists

PGPH-D-25-03067R1

Dear Dr. Talwar,

We are pleased to inform you that your manuscript 'The barriers and enablers of outbreak reporting in the Asia-Pacific region: a mixed methods study of field epidemiologists' has been provisionally accepted for publication in PLOS Global Public Health.

Best regards,

Roojin Habibi, Ph.D., M.Sc., J.D.

Academic Editor
